

# Adding Four Dimensional Data Assimilation by Analysis Nudging to the Model for Prediction Across Scales - Atmosphere (Version 4.0)

Orren Russell Bullock Jr.[1]., Hosein Foroutan[1,2], Robert C. Gilliam[1], Jerold A. Herwehe[1]

[1]Computational Exposure Division, National Exposure Research Laboratory, Office of Research and Development, U.S. Environmental Protection Agency, Research Triangle Park, North Carolina, USA
[2]Department of Biomedical Engineering, Virginia Polytechnic Institute and State University, Blacksburg, Virginia, USA

*Correspondence to*: Orren Russell Bullock Jr. (bullock.russell@epa.gov)

**Abstract.** The Model for Prediction Across Scales – Atmosphere (MPAS-A) has been modified to allow four dimensional data assimilation (FDDA) by the nudging of temperature, humidity and wind toward target values predefined on the MPAS-A computational mesh. The addition of nudging allows MPAS-A to be used as a global-scale meteorological driver for retrospective air quality modeling. The technique of "analysis nudging" developed for the Penn State / NCAR Mesoscale Model, and later applied in the Weather Research and Forecasting model, is implemented in MPAS-A with adaptations for its unstructured Voronoi mesh. Reference fields generated from $1° \times 1°$ National Centers for Environmental Prediction FNL (Final) Operational Global Analysis data were used to constrain MPAS-A simulations on a 92-25 km variable-resolution mesh with refinement centered over the contiguous United States. Test simulations were conducted for January and July 2013 with and without FDDA, and compared to reference fields and near-surface meteorological observations. The results demonstrate that MPAS-A with analysis nudging has high fidelity to the reference data while still maintaining conservation of mass as in the unmodified model. The results also show that application of FDDA constrains model errors relative to 2 m temperature, 2 m water vapor mixing ratio, and 10 m wind speed such that they continue to be at or below the magnitudes found at the start of each test period.

## 1 Introduction

Combining data at various times in a dynamical model to provide time continuity and dynamic balance among the model fields was first suggested by Charney et al. (1969) and has become known as four-dimensional data assimilation (FDDA). The use of special terms in forecast equations to "nudge" an atmospheric model simulation toward observations was originally employed for dynamic initialization (Anthes, 1974; Hoke and Anthes, 1976). Nudging was tested as a means for improving diagnostic simulations by Stauffer and Seaman (1990) in the Penn State / National Center for Atmospheric Research (NCAR) Mesoscale Model – Version 4 (MM4) (Anthes et al., 1987). In that study, nudging was applied in two ways. The model solutions were nudged toward either gridded analyses ("analysis nudging") or individual observations ("obs nudging"). After MM4, both forms of nudging were implemented in its successor MM5 (Grell et al., 1995) and subsequently in the Weather Research and Forecasting model (WRF) (Skamarock and Klemp, 2008).



In addition to basic variables like temperature, humidity, wind, cloud cover, and precipitation, meteorological simulations guided by nudging provide factors critical to air quality modeling that are not easily observed, such as stability, turbulence, mixing height, etc. Nudging applied in MM4, MM5 and WRF has been used at the U. S. Environmental Protection Agency (U.S. EPA) for almost three decades to support air quality modeling, first with the Regional Acid Deposition Model (RADM)

(Chang et al., 1987) and continuing to the present day with the Community Multi-Scale Air Quality (CMAQ) model (Byun and Schere, 2006; Appel et al., 2017).

It has long been recognized that air quality at any particular location can be affected by pollution sources on local to global scales (NRC, 1998; NRC, 2010). Air quality models are often applied with relatively coarse horizontal resolution on hemispheric and global scales to provide boundary information for nested, higher-resolution regional models (Bullock et al.,

2008; Jacobson and Ginnebaugh, 2010; Schere et al., 2012; Mathur et al., 2014). In various applications, this nested modeling strategy has created unrealistic simulations at the lateral boundaries of internal model domains due to discontinuities in horizontal and vertical resolution and/or differing modeling assumptions between separate models used at each scale (Warner et al., 1997; Bullock et al., 2009; Tudor and Termonia, 2010; Mathur et al., 2017).

To address the need for a global-to-local air quality modeling system that can avoid boundary problems associated with model

domain nesting, this work adapts the Model for Prediction Across Scales - Atmosphere (MPAS-A) (Skamarock et al., 2012) for use as the meteorological component of a future coupled meteorological-chemical modeling system. MPAS-A, which features a global unstructured Voronoi mesh, offers gradual mesh refinement rather than discrete nesting to a focal region. For retrospective air quality modeling, an FDDA approach based on analysis nudging has been developed and tested in MPAS-A as described below.

**2. Experimental design, implementation and testing**

FDDA by way of analysis nudging, similar to that described in Stauffer and Seaman (1990), has been added to MPAS-A. Unlike MM4, MM5 and WRF, which are limited-area models with rectangular computational grids, MPAS-A has an unstructured computational mesh as illustrated in Fig. 1. Nonetheless, once the required "target" fields (i.e., reference data for nudging) are defined to match the MPAS-A prognostic variable array, analysis nudging in MPAS-A is similar to its ancestral

implementations.

**2.1 Description of analysis nudging in MPAS-A**

Analysis nudging is applied in MPAS-A by the addition of a nudging tendency term to the normal predictive equation. The nudging tendency for prognostic variable α is calculated as:

$$\left(\frac{\partial \alpha}{\partial t}\right)_{nudge} = G_\alpha W_{PBL} W_{layer} \left(\alpha_{target} - \alpha\right),$$  (1)





where $G_\alpha$ is a nudging inverse time scale or "nudging coefficient", $W_{PBL}$ and $W_{layer}$ are special binary terms (1 or 0), and $\alpha_{target}$ is the target or reference value for $\alpha$. Trusted reference fields are typically only available at certain times each day and temporal interpolation is required to provide target values at each model time step. It may be advantageous to avoid nudging in the planetary boundary layer (PBL) so as not to disrupt simulated diurnal processes. In this case, $W_{PBL}$ can be set equal to 1 in

layers above the simulated PBL top and set equal to 0 in layers below or containing the PBL top. Otherwise $W_{PBL}$ can be set equal to 1 in all layers. $W_{layer}$ is a similar binary term to allow the exclusion of nudging near the surface based simply on vertical layer number. $G_\alpha$, $W_{PBL}$ and $W_{layer}$ are all defined independently for each of the nudged variables.

Analysis nudging has been applied for potential temperature ($\Theta$), water vapor mixing ratio ($q_v$), and wind. Treating wind involves extra complications because of the way it is represented in the unstructured mesh. As illustrated by Fig. 1, scalar

prognostic variables including $\Theta$ and $q_v$ are defined at the cell centers. However, the prognostic variable for wind in MPAS-A is the component perpendicular to the cell faces ($U$). To nudge wind, meridional and zonal decompositions at the cell centers are used. These model variables *UReconstructZonal* and *UReconstructMeridional* already exist to treat the influence of parameterized convection and PBL processes on the wind field. While the wind component across cell edges ($U$) could be nudged directly, this method would require 50% more comparisons between prognostic and target values since there are 3

15   times as many cell edges as there are cells. Nudging tendencies for *UReconstructZonal* and *UReconstructMeridional* are translated to cell edges in the same manner as the tendencies for PBL and convection processes.

## 2.2 Creating target fields

The MPAS-A modeling system already provides model initialization software, namely the executable program *init_atmosphere_model*. For this study, initialization fields were created at each time where nudging target fields were desired

using 1° x 1° NCEP FNL (Final) Operational Global Analysis data (ds083.2) (NCEP/NWS/NOAA/U.S. Department of Commerce, 2000). Target fields could be based on other analytical methods such as 3- and 4-dimensional variational assimilation (3D-VAR, 4D-VAR) or an ensemble Kalman filter (EnKF). However, the NCEP FNL data are already produced using techniques similar to 4D-VAR. The nudged variables $\Theta$, $q_v$, *UReconstructZonal* and *UReconstructMeridional* were extracted from each initialization file, renamed *th_fdda_new*, *qv_fdda_new*, *u_fdda_new*, and *v_fdda_new*, respectively, and

used to compile the necessary FDDA input files. The modified MPAS-A reads in new FDDA targets every 6 h when NCEP FNL data are available, specifically at 00, 06, 12, and 18 UTC. Target values at intervening times during the simulation are computed using linear time interpolation.

The FDDA target variable names contain "*new*" to indicate that, for the time increment at which they are read, the values represent the target value at the end of the upcoming 6-h FDDA time interval. Unlike WRF which reads in "old" and "new"

targets for each FDDA interval, the modified MPAS-A reads only "new" values. The "new" values from the previous time interval are recycled to be used as the "old" values, thus reducing the FDDA target file size by half. At simulation start time, initial values for $\Theta$, $q_v$, *UReconstructZonal* and *UReconstructMeridional* are used to set *th_fdda_old*, *qv_fdda_old*, *u_fdda_old*,



and *v_fdda_old*, respectively. Thus, simulation start time must be at 00, 06, 12, or 18 UTC in order to maintain the 6-h FDDA data interval.

Scripts have been written to automate the process of running *init_atmosphere_model* for each FDDA time, extracting the four nudged variables, and composing the FDDA target input file. They perform variable extraction and FDDA input file composition using NetCDF Operators (NCO) software available at http://nco.sourceforge.net/.

## 2.3 FDDA test applications

MPAS-A version 4.0 (https://github.com/MPAS-Dev/MPAS-Release/releases/tag/v4.0), modified to include FDDA by analysis nudging as described above, was applied on a 92-25km variable-resolution mesh obtained from the MPAS Atmosphere Public Releases web page (http://mpas-dev.github.io/atmosphere/atmosphere-download.html) with the origin of this mesh repositioned to 40° N, 95° W. Two test simulation periods were defined spanning January 2013 and July 2013.

As mentioned before, model initialization and FDDA inputs were produced from 1° x 1° NCEP FNL data using the *init_atmosphere_model* software included in the MPAS-A version 4.0 public distribution. For this study, *init_atmosphere_model* was slightly modified to allow finer vertical resolution near the surface. Air-quality models typically require fine vertical resolution in the PBL in order to better simulate pollutant emissions which are commonly near the surface. To produce sufficiently thin layers near the surface, the unmodified *init_atmosphere_model* required an unreasonable number of layers due to the 1.5-power function used to define layer boundary heights. Modified Fortran code described in Appendix A was developed such that only 50 layers were required with the model top specified at 30 km. Using the modified code and given a surface elevation at sea level, layer thickness is 18 m at the bottom, 232 m at 1.5 km elevation, 1000 m at 12.5 km elevation, and 1729 m at the top. This layer structure was used to produce model initialization and FDDA target data files for all MPAS-A model simulations described below.

The modified *init_atmosphere_model* was also used to produce update fields for sea surface temperature and sea ice at 6-h intervals throughout each test period. For this purpose, the new layer generation function had no bearing, but a problem was discovered in the original MPAS-A model code where sea ice was being analyzed over land areas. This problem was solved with additional code modifications described in Appendix B.

Once the required initialization, surface update and FDDA target fields were in place, MPAS-A simulations were performed with the *atmosphere_model* program. Table 1 shows all non-default *nhyd_model* and *damping* namelist options used in this study. Namelist options for *atmosphere_model* from the standard MPAS-A and their default values are described in Appendix B of the MPAS Atmosphere Model User's Guide version 4.0 (available as of 19 July 2017 at http://www2.mmm.ucar.edu/projects/mpas/mpas_atmosphere_users_guide_4.0.pdf). Table 2 shows all applicable *physics* namelist options chosen from the standard MPAS list. These do not include namelist options added to MPAS-A as part of the FDDA implementation.

Table 3 shows the new namelist options added as part of the FDDA implementation and the values used for testing in this study. These options are similar to those used in the WRF model for FDDA application. WRF contains them within a special





*fdda* subset of namelist options. For now, they have been added to the *physics* namelist input variable list for MPAS-A. The primary option to invoke FDDA is *config_fdda_scheme*. As a default, *config_fdda_scheme = off*, and FDDA is not invoked in the modified MPAS-A model. If FDDA is invoked with a value of *analysis*, then the other options in Table 3 become applicable. The modified MPAS-A code also includes a second option for FDDA called *scaled* which allows the user to adjust

nudging strength based on MPAS cell size. This option is still under development and has not been investigated as a part of this study.

FDDA can be selectively applied or omitted for each meteorological variable ($\Theta$, $q_v$, $U$). If applied, the nudging strength is controlled by a variable-specific nudging coefficient. FDDA for each variable can be applied throughout all vertical layers, or only above a particular layer number. In many previous applications of WRF, it was common for FDDA to only be applied

above the PBL so as not to disrupt the diurnal evolution of the PBL with data from a linear time interpolation (Otte et al., 2012; Bowden et al., 2012; Bowden et al., 2013; Bullock et al., 2014). Table 3 shows namelist options provided to avoid nudging in the PBL, avoid nudging below a specified layer number, or both. It is also worth noting here that the default nudging coefficients implemented in MPAS-A are equal to $3.0\times10^{-4}$ s$^{-1}$ for all variables, just like in WRF. Unlike for temperature and wind, nudging water vapor concentration perturbs atmospheric mass in the simulation. Previous studies using WRF have

chosen to employ smaller nudging coefficients for $q_v$ versus other variables (Otte et al., 2012; Bowden et al., 2012; Bowden et al., 2013; Bullock et al., 2014). Results discussed later in this work show some benefit from doing so. For this study, the nudging coefficient for $q_v$ was one order of magnitude smaller than for the other variables, except for a special test where the value was kept equal.

## 3. Results

To evaluate FDDA in MPAS-A, test simulations for January and July of 2013 were performed with both the standard version of the model and the modified model using FDDA by analysis nudging. The modified MPAS-A was also applied with FDDA turned off to verify agreement with the results obtained from the standard model. Model results from the standard and modified versions were first compared to the FDDA target fields. Obviously, nudging strongly toward the target fields should produce good agreement with those fields. The intent of these first comparisons was to verify that using nudging coefficients for

temperature, humidity and wind similar to those used in WRF would constrain MPAS-A simulations in a reasonable manner. To further test the capabilities of FDDA in MPAS-A, simulated surface-level data for temperature, humidity and wind speed from both the standard and modified MPAS-A were then compared to observational data from the Meteorological Assimilation Data Ingest System (MADIS) (https://madis.noaa.gov). Finally, total dry air, total water vapor and total atmospheric mass calculations were performed to test for any corruption of mass conservation by the implementation of FDDA in MPAS-A.





### 3.1 Comparisons to FDDA target fields

Figure 2 shows MPAS-A simulation results and FDDA target fields for potential temperature in layers 1, 28, and 45, for 00 UTC 11 January 2013, ten days into the simulation. The left column of maps shows layer 1 values from the standard MPAS-A (top), the FDDA target field (middle) and MPAS-A with FDDA applied (bottom). The center and right columns shows the

5 same information for layers 28 and 45, respectively. Layer 1 extends from 0 to ~18 m above the surface where the surface is at mean sea level (msl), and to ~15 m above the surface over the highest resolved terrain. The vertical span of layer 28 varies from 5002-5551 m to 9449-9784 m above msl depending on the resolved terrain height which varies from -82 to 5425 m. So layer 28 represents a 330-550 m thick layer somewhere in the middle troposphere. The span of layer 45 varies from 20622-22024 m to 20682-22048 m above msl. Layer 45 varies only slightly due to the MPAS-A hybrid vertical coordinate system

having shifted almost completely to a height coordinate at that altitude in the lower stratosphere.

By ten days into the simulation, the simulation without FDDA already shows significant potential temperature differences from the FDDA target fields in all three layers shown in Fig. 2. These differences are especially noticeable in layer 45 where an apparent stratospheric warming event is stronger in the "No FDDA" simulation and is longitudinally displaced about 120 degrees from the location in the reanalysis-based target field. The unconstrained simulation also shows a high-latitude cold

pool that is not in the target field and much colder stratospheric temperatures over the tropics. There are also some interesting differences in layer 28 around the high terrain of the Himalayas where the "No FDDA" simulation resulted in much warmer temperatures than the target values. The simulation with FDDA matches the target fields for $\Theta$ almost perfectly for layers 28 and 45. Near the surface, maps for layer 1 show difference from the $\Theta$ target field in both MPAS-A simulations, mostly in arctic regions where simulated temperatures are generally colder. But the simulation with FDDA shows surface temperatures

closer to the target values even though FDDA was not applied in the PBL.

To demonstrate that FDDA continues to constrain MPAS-A simulations through longer time periods, Fig. 3 shows the same information as Fig. 2, but this time the $\Theta$ fields are for 00 UTC 31 January 2013, 30 days into the simulation. The deviation of the "No FDDA" simulation from the target fields is larger than at day 10, but the results from the simulation with FDDA continue to follow the target fields closely for layers 28 and 45. However, layer 1 continues to be too cold across the arctic

with FDDA applied above the PBL. The simulation without FDDA is too cold in some parts of the arctic and too warm in others, and significant deviations from the target field are apparent in many locations around the globe. A quick investigation of observed surface temperatures at Barrow, Alaska, found that the simulated surface temperatures are about 10 K too cold at that location in the FDDA simulation. This points to the fact that FDDA applied only above the PBL can keep simulated temperatures at the surface from being too high due to the effect of convective mixing, but it cannot prevent them from being

too cold. Work is ongoing to remedy both warm and cold biases in surface temperature through the use of other land surface and PBL models which nudge soil temperature and moisture towards known conditions.

The array of maps in Fig. 4 shows water vapor mixing ratio at 00 UTC 31 January 2013 in the same arrangement as for potential temperature in Fig. 3. Even with weak $q_v$ nudging, the simulation with FDDA matches the target fields well for all three layers.





Without FDDA, the simulated pattern of water vapor deviates significantly from the target in all layers. In layer 45, the "No FDDA" case shows higher $q_v$ values all across the tropics than exist in the target field or in the simulation with FDDA where water vapor is practically absent. These results suggest that even weak nudging if water vapor can mitigate what appears to be artificial vertical diffusion of tropospheric water vapor into the stratosphere.

Fig. 5 shows layer 1 fields at 00 UTC 31 January 2013 for potential temperature and water vapor mixing ratio, but this time focused on the contiguous United States (CONUS). At this point in time, a strong cold front stretching from western Pennsylvania, down the Appalachian Mountains, and into the Gulf of Mexico was advancing from the west. It is important to note once again that analysis nudging was only applied above the PBL. Nonetheless, not only does the simulation with FDDA place the front in the correct location, but the simulated front shows sharper detail for $\Theta$ and $q_v$ than in the target fields. This

is especially true for $q_v$ where it appears that weaker nudging above the PBL allows simulated details at the surface to be better conserved. Finer detail is evident, not only along the cold front, but also in other locations where it appears variations in terrain and land cover type may be important. It stands to reason that if nudging had been applied within the PBL, these simulated fine details at the surface would have been blurred somewhat by blending with the less resolved target fields. Of course, finer detail does not in itself indicate better accuracy. To address that issue, simulation results were also compared to observational

data as described later in section 3b.

Vertical cross section plots of water vapor mixing ratio show that FDDA can constrain undesirable model behavior even when the nudging strength is quite weak. Figure 6 shows vertical cross sections along longitude 80° E from 55° N to 55° S for 00 UTC 31 January 2013 (day 30). This particular location was chosen to investigate the effect of the Himalayan Mountains on MPAS-A simulation results. The height of the Himalayan Mountains (~ 5 km) as resolved by the 92-km mesh size is shown

by the white area at the bottom of each plot. Figure 6a shows the standard MPAS simulation result, Fig. 6b shows the FDDA target field, and Fig. 6c shows the results from MPAS with FDDA using moisture nudging at one-tenth the strength used for the other variables. As previously seen in Fig. 4, the unconstrained simulation shows signs of upward transport and/or diffusion of water vapor into the lower stratosphere with water vapor concentrations over 2 orders of magnitude higher than the target values. The simulation with FDDA almost completely eliminates this deviation from the target, even with the weak nudging

strength.

Wind velocities and flow patterns from MPAS-A simulations and FDDA target fields were investigated with streamline plots. Figure 7 shows global streamline analyses for layer 28 (~500-300 hPa), again for 00 UTC 31 January 2013. As might be expected from a 30-day forecast, the flow field from the simulation without FDDA (Fig. 7a) differs significantly from the FDDA target flow field (Fig. 7b). However, the simulation with FDDA (Fig. 7c) follows the FDDA target data almost

perfectly. To show the ability of FDDA to maintain finer-scale fidelity, similar streamline analyses focused on the CONUS are shown in Fig. 8. Again, the simulation with FDDA is almost identical to the target flow field. Streamline analysis for layer 1 focused on the southeastern U.S. (Fig. 9) shows some noticeable differences between the simulation with FDDA and the target field. These differences are not surprising given that no nudging was applied in the PBL. Also, the FDDA target





fields above the PBL were derived from 1-degree FNL reanalysis data, while the simulation cell size is ~25 km in this region. Terrain effects on wind flow direction and speed appear to be more significant in the simulation than in the FDDA target field. Similar comparisons of MPAS-A simulations to FDDA target data were made for the July 2013 test period. All of these comparisons showed essentially the same results as were found for January 2013. While weather systems and patterns were

5 generally more quiescent, at least in the Northern Hemisphere, simulations unconstrained by FDDA still deviated significantly from target fields after a few days, while those constrained by FDDA maintained their fidelity relative to the target data.

### 3.2 Comparisons to observational data

MPAS-A simulation results were compared to observations of 2-m temperature, 2-m humidity and 10-m wind speed. To assure data quality, only aviation routine weather reports (METAR) and surface airways observation (SAO) reports from the

10 MADIS repository were used. This comparison was made using the Atmospheric Model Evaluation Tool (AMET) described in Appel et al. (2011). AMET was configured to calculate daily evaluation statistics for the entire global domain and for a sub-domain confined within 25 to 50 degrees North latitude and 67 to 125 degrees West longitude, basically covering the CONUS where the horizontal mesh size was 25 km. Daily statistics were calculated for both the January 2013 and July 2013 test periods.

Figure 10 shows the time series of 2-m temperature root mean squared error (RMSE) for January 2013. The top graph shows results for the entire global domain while the bottom graph shows results for the CONUS sub-domain. Three MPAS-A simulations were analyzed, the standard model denoted as *No FDDA*, FDDA applied using relatively weak $q_v$ nudging denoted as *FDDA*, and FDDA applied using equal nudging strength for all variables denoted as *FDDA (equal)*. The *No FDDA* cases show error increasing right from the start, both globally and over the CONUS. For the global domain, the *FDDA* and *FDDA*

*(equal)* cases both show RMSE actually decreasing somewhat over the first 10 days, but generally holding steady throughout the month. Globally, model performance with weak $q_v$ nudging is slightly better than with equal nudging strength for $\Theta$, $q_v$, and $U$. In the CONUS analysis, 2-m temperature RMSE for the *FDDA* case decreases more significantly over the first 10 days than in the global analysis, and remains below the starting values throughout the remainder of the month. Once again, relatively weak $q_v$ nudging improves model error statistics to some degree.

Figure 11 shows the same information as Fig. 10, except this time for July 2013. RMSE values are generally lower than for January 2013, but the same relationships hold between the *No FDDA* case and the other two cases. Slight reductions in 2-m temperature RMSE result from the use of weaker $q_v$ nudging.

Figures 12 and 13 show RMSE for 2-m humidity ($q_v$) during January 2013 and July 2013, respectively. These figures show RMSE values for July 2013 are much larger than for January 2013. This is largely due to the fact that MADIS observations

are more concentrated in the northern hemisphere. In fact, they are most concentrated in North America. Thus, in the northern hemisphere warm season when humidity levels are highest, model errors are also highest. But these figures show for humidity much the same effect of FDDA as was shown for temperature. Without FDDA, model error immediately increases at the start of the simulation and continues to increase for 10 or more days until errors in the unconstrained simulation approach the levels



of variation in the actual meteorological fields. From that time on, the magnitude of daily RMSE values fluctuate quite randomly. It is interesting to note in both Figs. 12 and 13 that the simulation with weaker $q_v$ nudging often has slightly lower RMSE than the simulation with equal nudging strength. While the difference is quite small, it is counterintuitive nonetheless. Apparently the 1° x 1° NCEP FNL data used to create the FDDA targets, with is relatively coarse resolution compared to the

25-km MPAS-A mesh used over North America, can degrade the simulation in that area where the MADIS observations are most concentrated. Further study is underway to see if target fields derived from newly available 0.25° x 0.25° NCEP FNL data lead to the same behavior.

Figures 14 and 15 show RMSE for 10-m wind speed during January and July of 2013, respectively. As with temperature and humidity, model errors for wind speed begin to increase at the start of both simulations without FDDA and continue to increase

for about 10 days. After 10 days, fluctuations in wind speed error in the unconstrained simulations appear to be quite random. The simulations with FDDA, regardless of the nudging strength for $q_v$, continue to have about the same RMSA for wind speed throughout the month, be that January or July of 2013. For wind speed, the strength of $q_v$ nudging appears to have little effect on RMSE. Opposite to what was seen for humidity, the analyzed wind speed errors are largest in the northern hemisphere cold season. The concentration of MADIS observations in the northern hemisphere is once again likely an important factor in

this seasonal difference in wind speed RMSE magnitude.

Even though FDDA nudging was not applied within the PBL for any variable, the results above show model errors near the surface were constrained quite well, except where simulated surface temperatures were too cold. Also, near the surface is where finer horizontal resolution of the model relative to the FDDA target data source has its greatest effect. Further study is anticipated to better identify optimal FDDA nudging strengths for $\Theta$, $q_v$, and $U$ in MPAS-A, and to better understand the

vertical levels of the atmosphere where nudging should be applied.

### 3.3 Mass Conservation tests

In addition to the comparisons described above, the ability of MPAS-A to conserve simulated atmospheric mass was also tested. For each month-long test period, all model simulations reported total atmospheric mass and total water vapor mass at each 150-second simulation time step. This was accomplished with minor additions to the Fortran code in the time integration

module (./src/core_atmosphere/dynamics/mpas_atm_time_integration.F).

As shown in Fig. 16, all simulations including those with FDDA conserved total moist air within the global model domain to within five parts in 100,000 of their starting values. Total water vapor mass varied more significantly in time in each simulation, and this is to be expected due to evaporation and precipitation processes. There is a diurnal signal evident in the water vapor mass total from all simulations, most likely due to longitudinal variations in evaporation and precipitation potential

under solar radiation caused by the geographic distribution of continents and oceans. For January 2013. the *No FDDA* simulation lost over 5% of its initial quantity of water vapor. This could be indicative of too much simulated precipitation, too little simulated evaporation, or both. Also, the model initialization based on the NCEP FNL analysis could be too moist. The simulations with FDDA all tended to maintain more total water vapor relative to the standard model. In general, the





unconstrained simulations tended to lose water vapor at the start of the simulation and come to an equilibrium point significantly lower that the simulations with FDDA. The *FDDA (equal)* simulations tended to quickly establish and then maintain the most total water vapor. Obviously, the FNL analysis indicated a moister atmosphere than the unconstrained MPAS-A simulations could maintain.

5 It is interesting to note that the *FDDA* and *FDDA (equal)* cases had almost identical trends in total moist air mass, with any differences in total water vapor almost perfectly cancelled out by differences in dry air. Also, the January 2013 *No FDDA* simulation gained total dry air mass to about 13 parts in 100,000 of the initial value. This is the same simulation that lost a significant fraction of its initialized water vapor, once again showing an opposite conservation response between dry air and water vapor.

10 Overall, the results in Fig. 16 demonstrate that the addition of FDDA does not degrade mass conservation relative to the standard MPAS-A. Conservation of dry air mass is most important if MPAS-A is to be used as the meteorological driver for air quality modeling. These results show that using FDDA could actually offer some improvement in that regard.

## 4. Summary and conclusions

The U. S. EPA is working to make MPAS-A suitable for use as the meteorological component of an integrated meteorology 15 and air-quality modeling system for global-to-fine-scale applications. The ability to constrain simulated meteorology to resemble historical reanalysis fields at comparable spatial scales is crucial to making this integrated modeling system a practical diagnostic tool for air-quality research. FDDA applied through analysis nudging has been used for decades to provide this constraint in other models such as MM4, MM5 and WRF. The results shown here demonstrate that it also works quite well in MPAS-A. Comparison of MPAS-A simulations of January and July 2013 with and without FDDA demonstrate that 20 unconstrained simulations deviate significantly from historical conditions in only a few days, while those constrained through analysis nudging follow historical conditions well in most situations. Due to surface-layer decoupling, analysis nudging applied only above the PBL was not able to constrain the development of excessively cold surface temperatures in arctic areas during the January 2013 simulation period. However, this can be addressed with the use of land surface models that also employ FDDA. Further study is already underway at the U.S. EPA to determine the best strength with which to nudge 25 temperature, humidity and wind in MPAS-A and the levels of the atmosphere to best apply that nudging.

The target fields toward which MPAS-A state variables are nudged could come from a number of sources. Historical meteorological reanalysis products have previously been used for this purpose in regional and hemispheric modeling with WRF, and the results here suggest they can also be used with MPAS-A on the global scale. This study applied MPAS-A with a variable 92-25km mesh with the refined region centered on North America. Target fields used here were based on the 1° x 30 1° NCEP FNL reanalysis product. As such, there was not a great disparity in horizontal resolution between the simulations and the target fields where the MPAS-A mesh size was 92 km. However, where the mesh was more refined, MPAS-A was capable of delivering additional horizontal detail and the results shown here indicate weaker nudging may produce superior



results, at least when nudging water vapor mixing ratio. Finally, adding FDDA did not disrupt the ability of MPAS-A simulations to conserve mass, and this is an important point when considering its use for air-quality modeling.

**Code and data availability**

The MPAS-A model software used in this project is a subset of the complete Model for Prediction Across Scales (MPAS)
developed by Los Alamos National Security, LLC (LANS) and the University Corporation for Atmospheric Research (UCAR) and distributed under a 3-clause BSD license allowing distribution of original and derivative works under conditions that have been satisfied here. The full text of this BSD license can be found in http://mpas-dev.github.io/files/documents/MPAS-DevelopersGuide.pdf. MPAS-A model source codes used in this study are available in the Supplement and at https://doi.org/10.5281/zenodo.1101204 with all modified codes accompanied by their original codes. Run scripts used to
prepare FDDA target fields are also included in the Supplement and at https://doi.org/10.5281/zenodo.1101204. The definition file for the 92-25km computational mesh used in this study is too large for the Supplement, but it can be obtained from https://doi.org/10.5281/zenodo.1101204. Operational model global tropospheric analysis data used to initialize MPAS-A and to define FDDA target fields are available at https://rda.ucar.edu/datasets/ds083.2/. The Atmospheric Model Evaluation Tool used in this study is available at https://www.cmascenter.org/amet/. Observational data used within AMET were obtained
from https://madis.noaa.gov/.

**Appendix A**

The following describes FORTRAN code modifications made to define a vertical layer structure with fine resolution near the surface without the use of an extremely large number of layers. The basis for these changes was the MPAS-A model code originally published in the MPAS Version 4.0 code release dated 22 May 2015. These changes are included in the MPAS
model codes provided in the Supplement and at https://doi.org/10.5281/zenodo.1101204.

In src/core_init_atmosphere/mpas_init_atm_cases.F, make the following edits:

Replace line 2538 with the following:
real (kind=RKIND) :: r_earth, etavs, ztemp, zd, zt, gam, delt, str, grd, kfrac

Replace lines 2864 through 2870 with the following:
    write(0,*) '*** Using custom layer definition ***'
    str = 3.
grd = 0.03

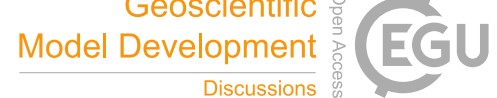

```
zt = config_ztop
do k=1,nz
  kfrac = real(k-1)/real(nz1)
  zw(k) = zt*((1-grd)*kfrac**str+grd*kfrac)
```

## 5 Appendix B

The following information was obtained from the MPAS GitHub web site to fix a problem with sea ice being defined over land areas. The actions described below have already been taken in the MPAS model codes provided in the Supplement and at https://doi.org/10.5281/zenodo.1101204.

As of 5 October 2017, the original MPAS GitHub information was available at the location shown below.

https://github.com/nickszap/MPAS-Release/commit/88f730142fc2ea04db12aa5e37f3337114e2ac45

Description: Activate the 'vertical_stage_in' package when config_init_case == 8

The init_atmosphere core previously only activated packages to read in static fields when config_init_case == 7 (and either config_vertical_grid or config_met_interp were true). However, for config_init_case == 8, we need the landmask field for

interpolating, e.g., sea-ice. Since we may potentially need other static fields as well when creating surface update files, the simplest solution seems to be to simply read all static fields when config_init_case == 8 by activating the vertical_stage_in package for this case.

Action:

This action was taken by adding the following lines of FORTRAN code after line 162 in src/core_init_atmosphere/mpas_init_atm_core_interface.F as originally published in the MPAS Version 4.0 code release dated 22 May 2015.

```
else if (config_init_case == 8) then
  vertical_stage_in = .true.
  vertical_stage_out = .false.
  met_stage_in = .false.
  met_stage_out = .false.
```

## Acknowledgements

This research was performed while Hosein Foroutan held a National Research Council Research Associateship Award at the U.S. EPA. We thank Tanya Spero and Jon Pleim (U.S. EPA) for their internal review of the manuscript and providing



constructive comments. The views expressed in this article are those of the authors and do not necessarily represent the views or policies of the United States Environmental Protection Agency.

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





Table 1. MPAS-A non-default namelist configuration variables used for testing (except physics)

*nhyd_model*

| config_dt | Model time step, in seconds<br>*Applied value: 150.0* |
|---|---|
| config_start_time | Starting time for model run<br>*Applied value: '2013-01-01_00:00:00' and '2013-07-01_00:00:00'* |
| config_run_duration | Length of model run<br>*Applied value: '31_00:00:00'* |
| config_len_disp | Horizontal length scale for Smagorinsky formulation of horizontal diffusion<br>*Applied value:  25000.0* |
| config_h_ScaleWithMesh | Scale eddy viscosities with mesh-density function for horizontal diffusion<br>*Applied value:  .true.* |

*damping*

| config_zd | Height MSL to begin w-damping profile<br>*Applied value: 27000.0* |
|---|---|
| config_xnutr | Maximum w-damping coefficient at model top<br>*Applied value: 0.2* |





Table 2.  Standard MPAS-A physics namelist variables used for testing

| | |
|---|---|
| config_sst_update | Logical used to update the Sea-Surface Temperatures (SSTs) and fractional sea-ice.  If set to true, SSTs are updated using the file config_sfc_update_name. If set to false, SSTs remain fixed during the entire model run. *Applied value: .true.* |
| config_sstdiurn_update | If set to true, a diurnal cycle is applied to the SSTs.  If set to false, SSTs remain constant during the entire day. *Applied value: .false. (same as default value)* |
| config_deepsoiltemp_update | If set to true, deep soil temperatures are slowly updated during the model run. If set to false, deep soil temperatures remain fixed during the entire run. *Applied value: .false. (same as default value)* |
| config_radtlw_interval | Temporal interval between calls to the parameterizations of long wave radiation, format *'yyyy-mm-dd_hh:mm:ss'*. *Applied value: '00:10:00'* |
| config_radtsw_interval | Temporal interval between calls to the parameterizations of short wave radiation, format *'yyyy-mm-dd_hh:mm:ss'*. *Applied value: '00:10:00'* |
| config_bucket_update | Temporal interval between updates to restoring the accumulated rain and radiation fields below their respective bucket values, format *'yyyy-mm-dd_hh:mm:ss'*. *Applied value: 'none' (same as default value)* |
| config_physics_suite | Physics suite: *Not applicable.* |
| config_microp_scheme | Cloud Microphysics scheme: *Applied value: 'wsm6'* |
| config_convection_scheme | Convection scheme: *Applied value: 'kain_fritsch'* |
| config_lsm_scheme | Land-surface scheme: *Applied value: 'noah'* |
| config_pbl_scheme | Planetary Boundary Layer scheme: *Applied value: 'ysu'* |
| config_radt_cld scheme | Parameterization of cloud fraction for long and short wave radiation schemes: *Applied value: 'cld_fraction'* |
| config_radt_lw_scheme | Long wave (LW) radiation scheme: *Applied value: 'rrtmg_lw'* |
| config_radt_sw_scheme | Short wave (SW) radiation scheme: *Applied value: 'rrtmg_sw'* |
| config_sfclayer_scheme | Surface-layer scheme: *Applied value: 'monin-obukhov'* |

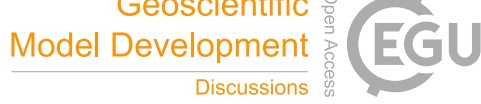

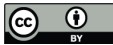

Table 3. New MPAS-A physics namelist variables added for the FDDA implementation and the values used for testing.

| | |
|---|---|
| config_fdda_scheme [*] | Four-dimensional data assimilation (FDDA) scheme:<br>'none' = FDDA not applied (default value)<br>'analysis' = analysis nudging with constant nudging strength<br>'scaled' = analysis nudging with scale-dependent nudging strength<br>*Applied value: 'analysis'* |
| config_fdda_t [*] | Potential temperature nudging indicator:<br>.true. = apply nudging to potential temperature<br>.false. = do not apply nudging to potential temperature (default value)<br>*Applied value: .true.* |
| config_fdda_q [*] | Water vapor mixing ratio nudging indicator:<br>.true. = apply nudging to water vapor mixing ratio<br>.false. = do not apply nudging to water vapor mixing ratio (default value)<br>*Applied value: .true.* |
| config_fdda_uv [*] | Wind nudging indicator:<br>.true. = apply nudging to wind<br>.false. = do not apply nudging to wind (default value)<br>*Applied value: .true.* |
| config_fdda_t_coef | Nudging coefficient for potential temperature ($s^{-1}$), default value = $3.0 \times 10^{-4}$.<br>*Applied value: $3.0 \times 10^{-4}$* |
| config_fdda_q_coef | Nudging coefficient for water vapor mixing ratio ($s^{-1}$), default value = $3.0 \times 10^{-4}$.<br>*Applied value: $3.0 \times 10^{-5}$(base case), $3.0 \times 10^{-4}$(sensitivity test)* |
| config_fdda_uv_coef | Nudging coefficient for wind ($s^{-1}$), default value = $3.0 \times 10^{-4}$.<br>*Applied value: $3.0 \times 10^{-4}$* |
| config_fdda_t_in_pbl [*] | If config_fdda_t = .true., nudge potential temperature in PBL?<br>.true. = yes (default value)<br>.false. = no<br>*Applied value: .false.* |
| config_fdda_q_in_pbl [*] | If config_fdda_q = .true., nudge water vapor missing ratio in PBL?<br>.true. = yes (default value)<br>.false. = no<br>*Applied value: .false.* |
| config_fdda_uv_in_pbl [*] | If config_fdda_uv = .true., nudge wind in PBL?<br>.true. = yes (default value)<br>.false. = no<br>*Applied value: .false.* |
| config_fdda_t_min_layer [*] | If config_fdda_t = .true., lowest layer to nudge potential temperature,<br>Default value = 0.<br>*Applied value: 0* |





| config_fdda_q_min_layer [*] | If config_fdda_q = .true., lowest layer to nudge water vapor mixing ratio, Default value = 0. *Applied value: 0* |
|---|---|
| config_fdda_uv_min_layer [*] | If config_fdda_uv = .true., lowest layer to nudge wind, Default value = 0. *Applied value: 0* |




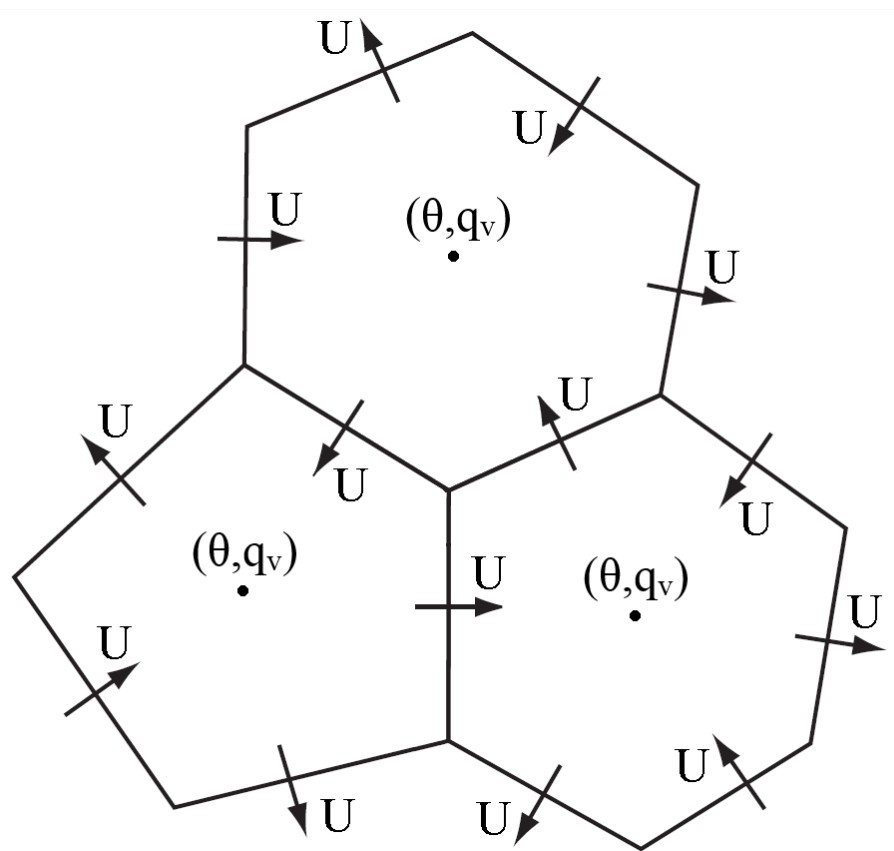

**Figure 1: A schematic of C-grid staggered variables on the MPAS-A horizontal mesh. Normal wind velocities (U) are defined on the cell faces while all other scalar variables are defined at the cell centers, including potential temperature (θ) and water vapor mixing ratio (qv).**



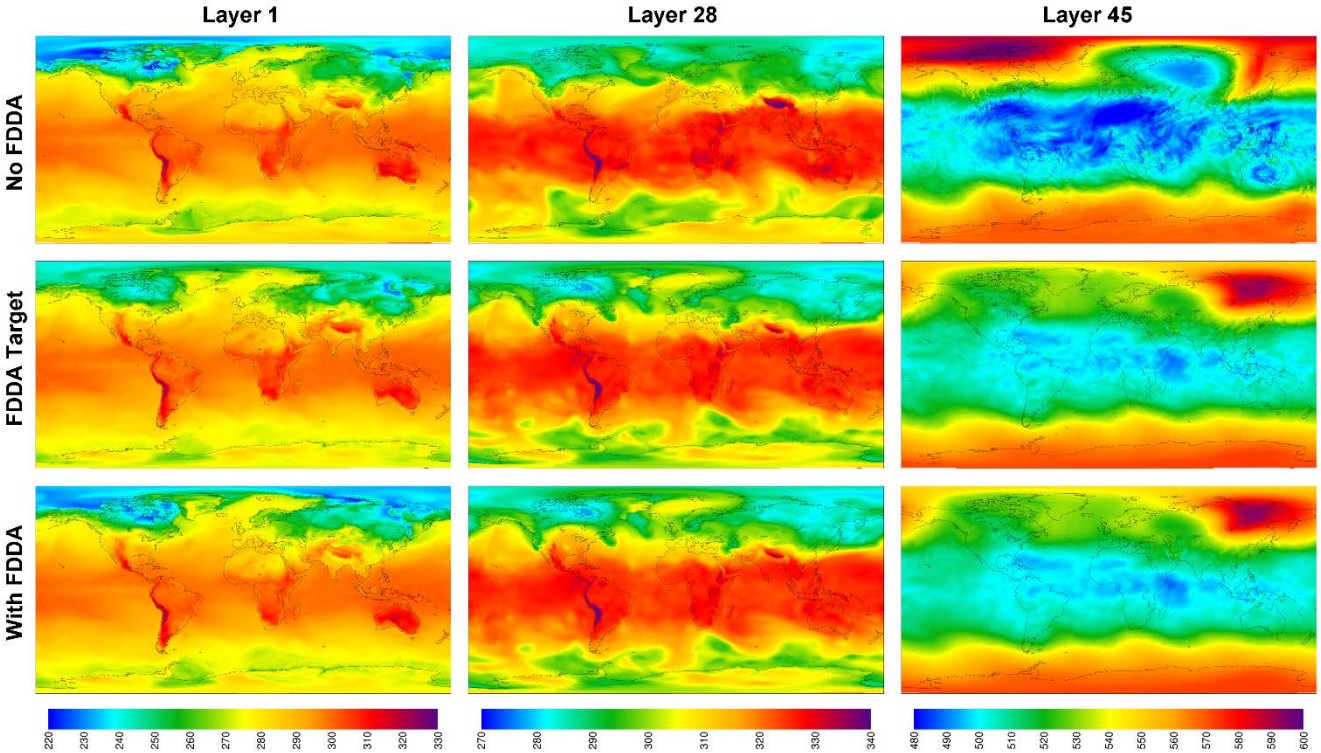

**Figure 2: MPAS-A simulation results without FDDA (top row), FDDA target fields (center row), and MPAS-A results with FDDA (bottom row) for potential temperature in layers 1, 28, and 45, for 00 UTC on 11 January 2013, 10 days into the simulation.**





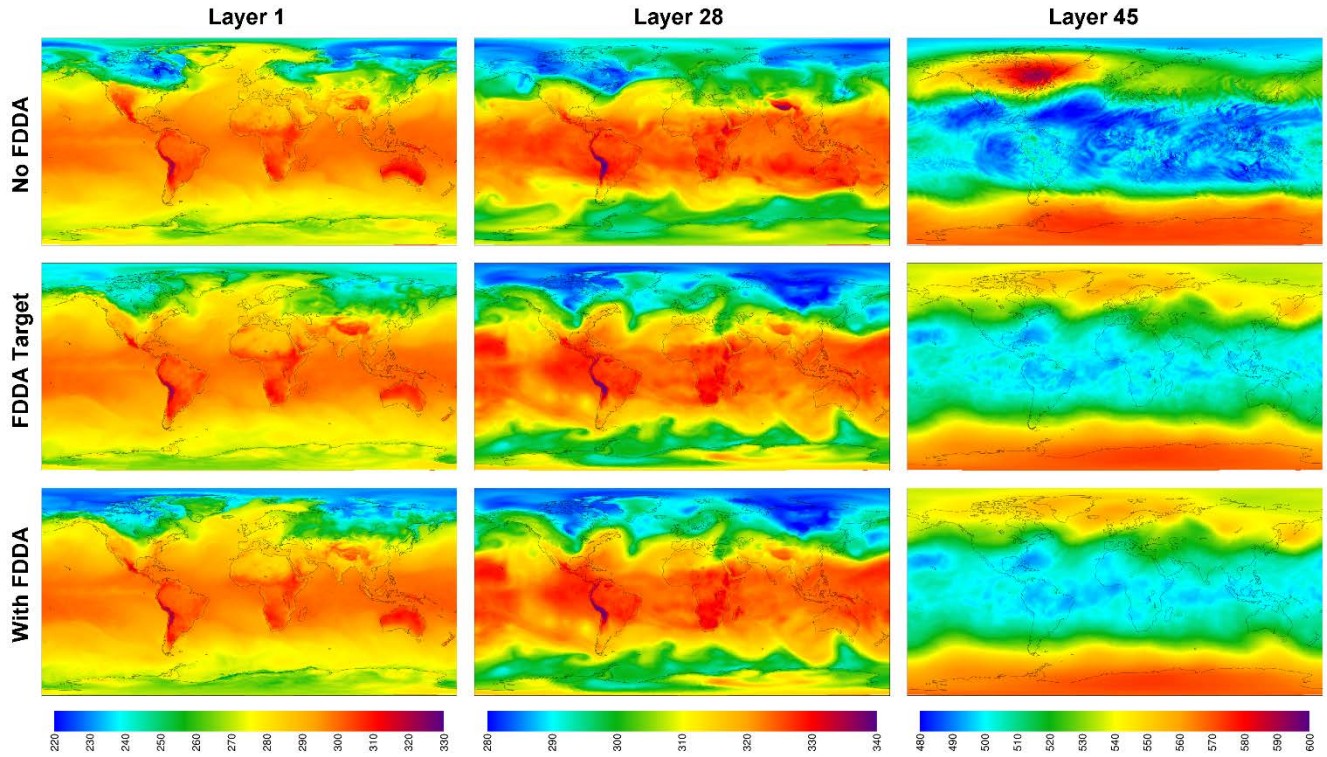

**Figure 3: MPAS-A simulation results without FDDA (top row), FDDA target fields (center row), and MPAS-A results with FDDA (bottom row) for potential temperature in layers 1, 28, and 45, for 00 UTC on 31 January 2013, 30 days into the simulation.**





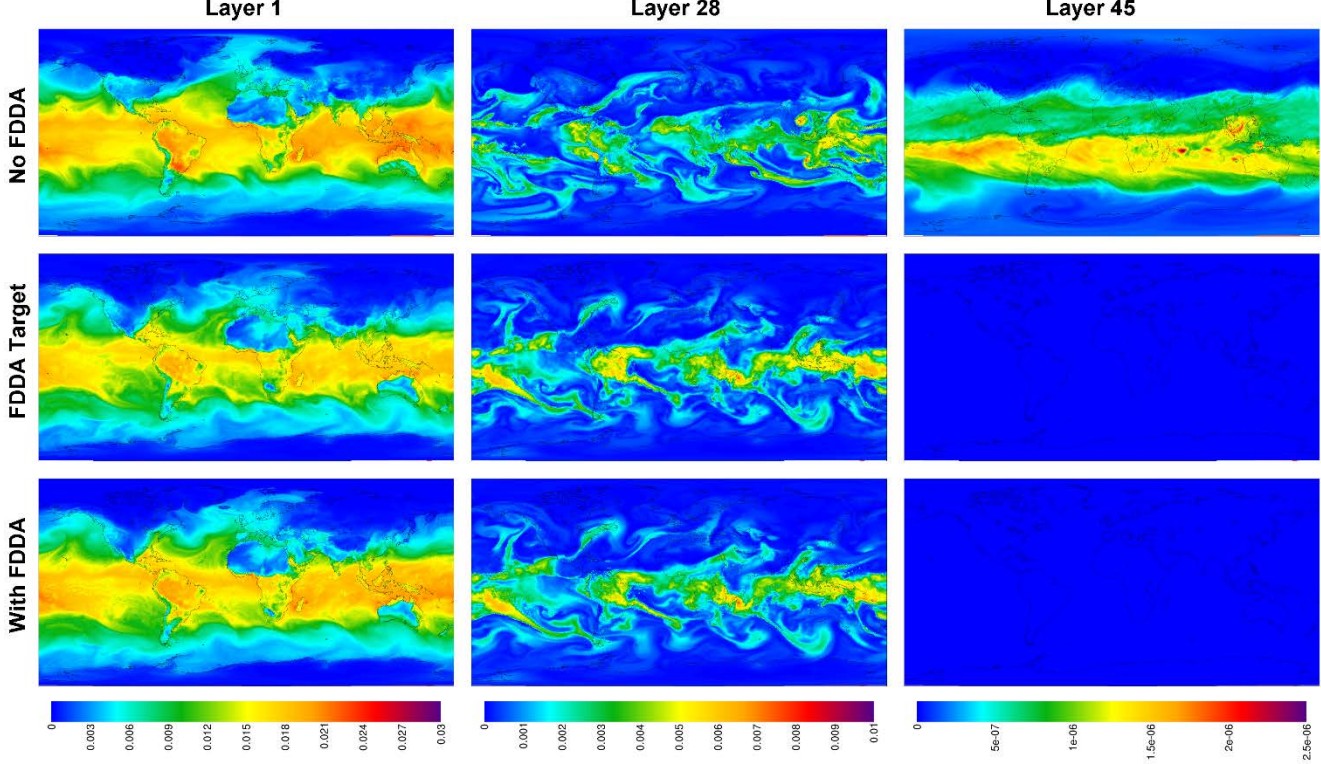

**Figure 4: MPAS-A simulation results without FDDA (top row), FDDA target fields (center row), and MPAS-A simulation results with FDDA (bottom row) for water vapor mixing ratio in layers 1, 28, and 45, for 00 UTC on 31 January 2013, 30 days into the simulation.**





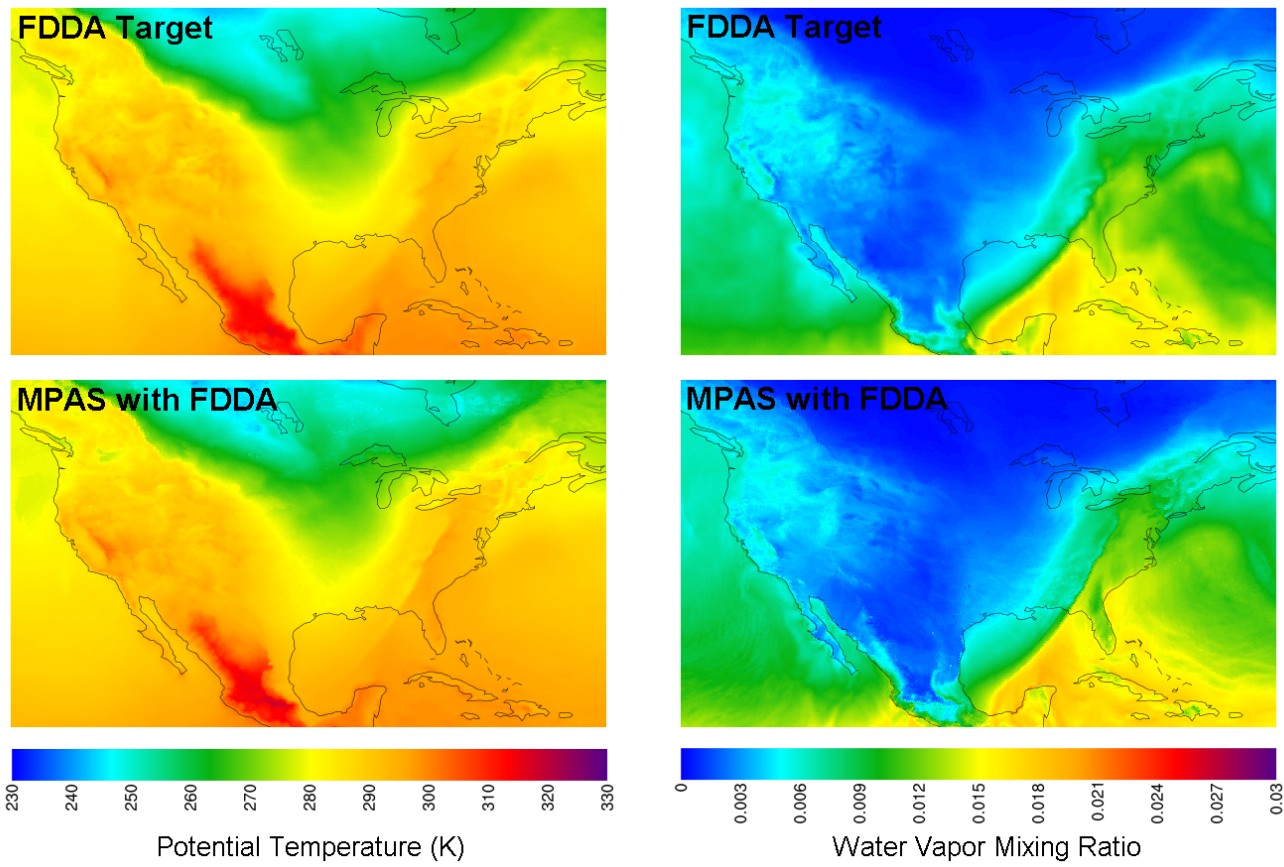

**Figure 5: Simulation results from MPAS-A using FDDA and FDDA target fields for potential temperature and water vapor mixing ratio at 00 UTC on 31 January 2013 focused on the contiguous United States.**





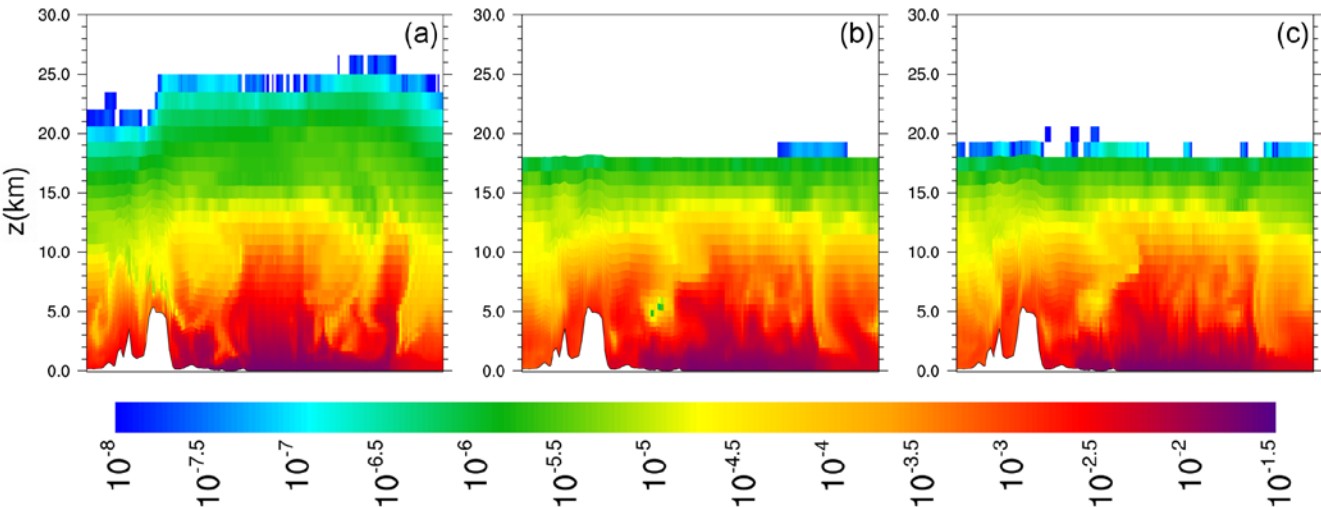

**Figure 6: Cross-sectional plots of water vapor mixing ratio along longitude 80° E from 55° N to 55° S for 00 UTC on 31 January 2013, (a) from MPAS-A without FDDA, (b) FDDA target field, and (c) from MPAS-A using FDDA at one-tenth strength compared to the other nudged variables**





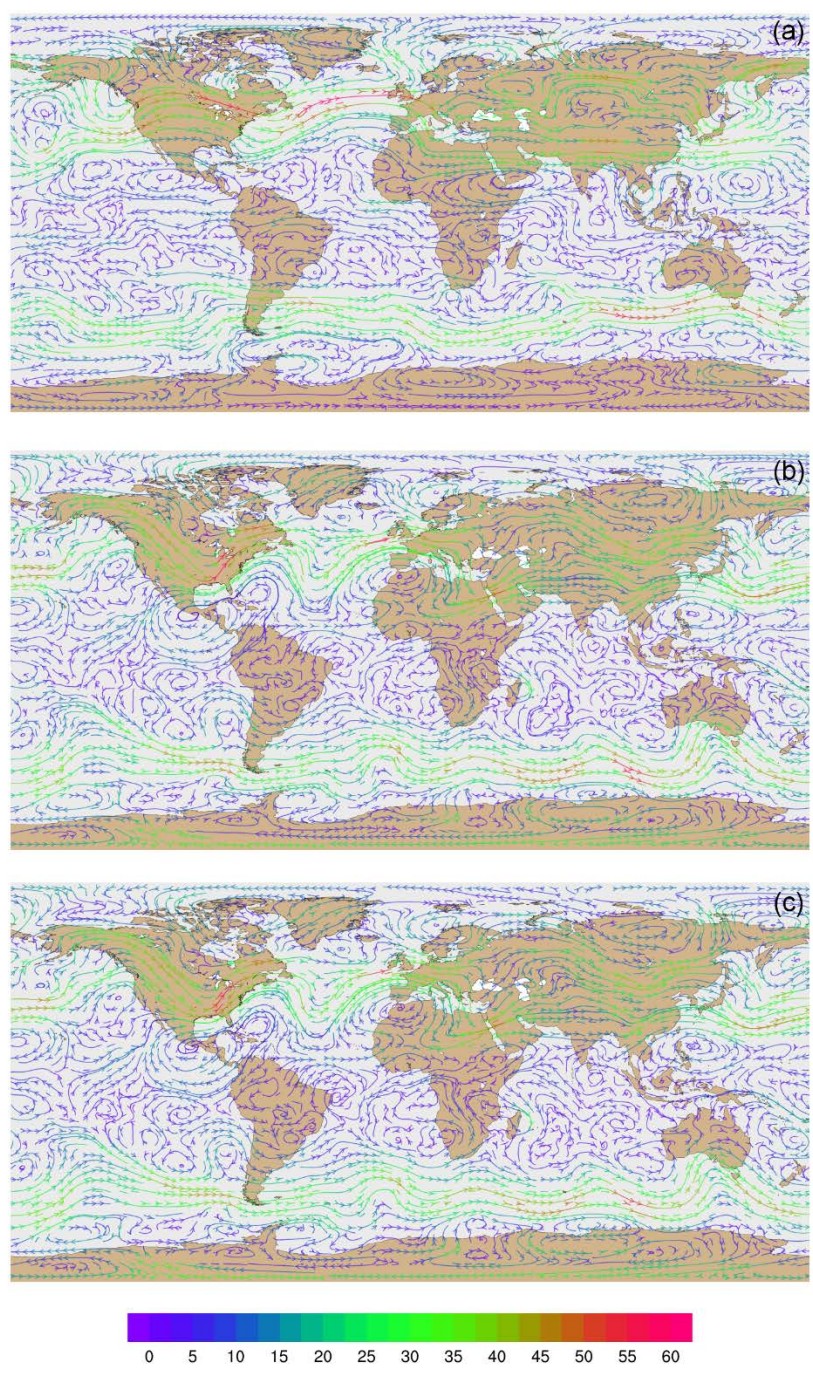

**Figure 7: Global streamline analyses for layer 28 (~500 300 hPa) at 00 UTC on 31 January 2013, (a) from MPAS-A without FDDA, (b) FDDA target field, and (c) from MPAS-A using FDDA. Streamline color indicates wind speed (m/s).**





**Figure 8: Streamline analyses for layer 28 (~500 300 hPa) at 00 UTC on 31 January 2013 focused on the contiguous United States, (a) from MPAS-A without FDDA, (b) FDDA target field, and (c) from MPAS-A using FDDA. Streamline color indicates wind speed (m/s).**



**Figure 9: Streamline analysis for layer 1 (surface) at 00 UTC on 31 January 2013 focused on the southeastern United States, (a) from MPAS-A using FDDA and (b) FDDA target field. Streamline color indicates wind speed (m/s).**





**Figure 10: January 2013 time-series plots of 2 m temperature root mean squared error (RMSE) for MPAS-A without FDDA (black line), MPAS-A with FDDA applied using weaker qv nudging (red line), and MPAS-A with FDDA applied at equal strength for all variables. The top graph shows results for the entire global domain while the bottom graph shows results for the CONUS sub-domain.**



**Figure 11: July 2013 time-series plots of 2 m temperature root mean squared error (RMSE) for MPAS-A without FDDA (black line), MPAS-A with FDDA applied using weaker qv nudging (red line), and MPAS-A with FDDA applied at equal strength for all variables. The top graph shows results for the entire global domain while the bottom graph shows results for the CONUS sub-domain.**





**Figure 12: January 2013 time-series plots of 2 m water vapor mixing ratio (qv) root mean squared error (RMSE) for MPAS-A without FDDA (black line), MPAS-A with FDDA applied using weaker qv nudging (red line), and MPAS-A with FDDA applied at equal strength for all variables. The top graph shows results for the entire global domain while the bottom graph shows results for the CONUS sub-domain.**





**Figure 13: July 2013 time-series plots of 2 m water vapor mixing ratio (qv) root mean squared error (RMSE) for MPAS-A without FDDA (black line), MPAS-A with FDDA applied using weaker qv nudging (red line), and MPAS-A with FDDA applied at equal strength for all variables. The top graph shows results for the entire global domain while the bottom graph shows results for the CONUS sub-domain.**





**Figure 14: January 2013 time-series plots of 10 m wind speed root mean squared error (RMSE) for MPAS-A without FDDA (black line), MPAS-A with FDDA applied using weaker qv nudging (red line), and MPAS-A with FDDA applied at equal strength for all variables. The top graph shows results for the entire global domain while the bottom graph shows results for the CONUS sub-domain.**





**Figure 15: July 2013 time-series plots of 10 m wind speed root mean squared error (RMSE) for MPAS-A without FDDA (black line), MPAS-A with FDDA applied using weaker qv nudging (red line), and MPAS-A with FDDA applied at equal strength for all variables. The top graph shows results for the entire global domain while the bottom graph shows results for the CONUS sub-domain.**





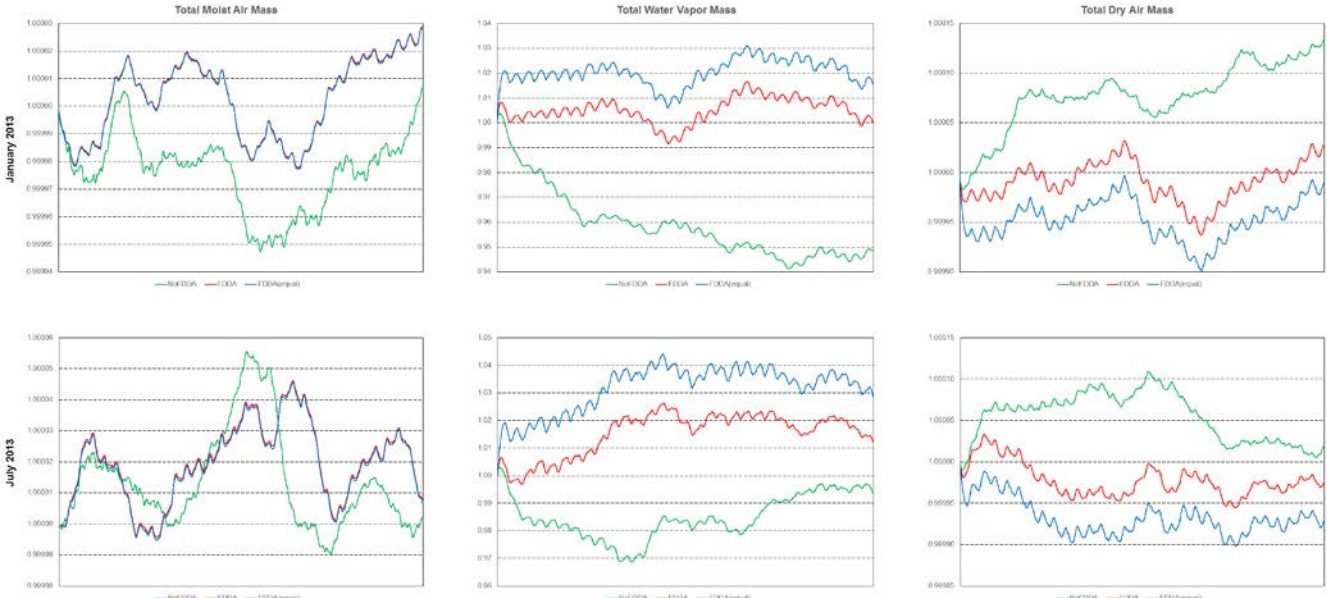

**Figure 16: Time-series plots of total moist air mass (left column), total water vapor mass (center column), and total dry air mass (right column) scaled to their initial values for the two test periods. Results are from MPAS A without FDDA (green line), MPAS-A using weaker qv nudging (red line), and MPAS A using equal nudging for all variables (blue line).**

