# Peer review of "Adding Four-Dimensional Data Assimilation by Analysis Nudging to the Model for Prediction Across Scales - Atmosphere (Version 4.0)"

_Geoscientific Model Development, 2017_

## Referee Comment (RC1) · Anonymous Referee #1 · 19 Feb 2018

General comments:

This paper looks at the use of nudging on a MPAS-A model using data from NCEP Final Analysis Data. It is a form of down-scaler, where additional value is being added by increases in vertical and horizontal resolution.

The paper is overall well written. It has scientific relevance.

I do have a few comments:

1) Biggest problem I see is that in Figures 10-14 – the RMSE for the NCEP FNL data on

its own without MPAS is not included. Is that because it is not a standard output within the NCEP FNL data? If it does exist it should be included in those plots. Otherwise we don't know whether we are actually getting additional value from this product. a. This issue needs to be resolved or at the very least explained before publication. Maybe the RMSEs need to be compared against those from the CMAQ model. b. If this paper is the first step to creating a competitive product it needs to say so.

The above issue is between what I would consider to be borderline between a minor revision and a major one.

I am struck by the gross similarity between FDDA target and MPAS with FDDA in Figures 2-8. It would be good to see plots where the differences between FDDA target and MPAS with FDDA were presented. Can you give some theoretical reasoning for choosing the time-scale forcing to be around 55 minutes – it seems to be a bit on the strong side. The weaker forcing for q is around 9 $\frac{1}{4}$ hours. Maybe this exists in the WRF version.

Specific comments:

Pg 3 Can the paper be a bit more specific on the details of how the NCEP FNL data is created?

Throughout the paper the resolution of the NCEP FNL data is described in terms of 1 degree resolution which is around 100km. Given that MPAS resolution is described in terms of km can you mention the FNL data resolution in km to avoid the readers needing to do the conversion themselves.

What is the timestep of the MPAS model?

Technical corrections:

Can you use Figure throughout and not have a mixture of Figure, Fig. and (Fig. 9) on pages 6-10.

Pg 9 line 11 RMSA -> RMSE Pg 9 line 30 remove dot : For January 2013

Pg 26 and Pg 27 Figures 7 and 8 need to include hyphen between 500 and 300 hPa

The plots in Figure 16 do not print out well. Can the units be included in the graph.

―――――――――――――――――――

---

## Referee Comment (RC2) · Anonymous Referee #2 · 9 Mar 2018

The manuscript "Adding Four Dimensional Data Assimilation by Analysis Nudging to the Model for Prediction Across Scales - Atmosphere (Version 4.0)" provides a description of the implementation of a nudging scheme into the MPAS model and provides results from an experiment testing this scheme by assimilating synthetic observations from a target field into the MPAS model state.

While the manuscript is well-written and provides a detailed description of the implementations, experiments and results, I find some major issues which should be addressed before publication (see comments below). Therefore, I recommend "major

revision" for the manuscript.

Major comments:

1) As the focus of the manuscript is nudging as a DA method, the authors should extend the introduction section with the focus being more on nudging and less on air quality.

2) The implementation of the nudging algorithm and the setup of the experiment is well documented in the manuscript. However, there are some issues with the experiment setup which in my opinion are important to discuss: a) The observations assimilated into the model state are synthetic observations generated from NCEP analysis fields. As the implementation does work as expected, the nudged simulations closely resemble the input data. However, one major advantage of the nudging algorithm is to allow for a free simulation the atmospheric processes corresponding to the model physics while being (temporally) constraint to the observations (or the target field in this case). With respect to the experiment design, the authors should add more discussion (and probably analysis) on this aspect. The experiment without PBL assimilation seems to be a natural starting point for this. b) The setup of the experiment is based on the truth being represented by the target field. The synthetic observations are drawn from this field in a homogeneous and dense fashion. With this setup, it is not difficult to achieve an accordance of the model with the target and the result will not strongly depend on the nudging coefficient(s). However, in reality, observations of the truth are heterogeneous, sparse and rare. Therefore, finding a reasonable and balanced setup which produces sound estimates of the atmospheric states is much harder especially with respect to the temporal availability of observations. Do the authors intend to do such experiments/simulations in the future? The authors should also provide results of sensitivity experiments with respect to the nudging coefficient.

3) The authors should provide plots containing information on the analysis increments from the nudging with respect to its spatial and temporal variability.

Minor comments:

Page 2 Line 17: Please provide a reference for the Voronoi mesh.

Page 2 Line 23: Is the mesh really unstructured?

Page 6 Line 4: "show" instead of "shows"

Page 7 Line 3f: I a not able to comprehend what the authors want to say with this sentence.

Page 8 Line 29: It would rather say "are larger" instead of "are much larger"

Page 9 Line 11: "RMSE" instead of "RMSA"

Page 10 Line 2: "than" instead of "that"

Discussion on Figures 10 to 13: The shape of the bottom and top diagram differ mostly by amplitude. Does this behavior arise from the fact that a majority of observations is coming from the CONUS region? Please elaborate on this.

---

## Author Comment (AC1) · 6 Apr 2018

Comment: Biggest problem I see is that in Figures 10-14 – the RMSE for the NCEP FNL data on its own without MPAS is not included. Is that because it is not a standard output within the NCEP FNL data? If it does exist it should be included in those plots. Otherwise we don't know whether we are actually getting additional value from this product. a. This issue needs to be resolved or at the very least explained before publication. Maybe the RMSEs need to be compared against those from the CMAQ model. b. If this paper is the first step to creating a competitive product it needs to say

so.

Response: While surface values for 2-meter temperature, 2-meter water vapor and 10-m wind are included in the NCEP FNL product, these data are not used in the generation of the FDDA target fields. As with analysis nudging in WRF, the analysis nudging we have developed for MPAS only uses target values defined on vertical layers within the model's computational grid. Also, the NCEP FNL data are only available at 6-hour intervals. But to satisfy our curiosity arising from this comment, we evaluated the NCEP FNL surface data against hourly METAR observations in the CONUS region using linear time interpolation to get the same hourly time resolution we used for the MPAS-A evaluation. The results showed superior accuracy compared to our MPAS simulations. Since METAR observations are used directly in the generation of the NCEP FNL surface values, this finding of superior accuracy is not surprising and comparison of MPAS-A and NCEP FNL regarding their agreement with METAR data is not really a fair assessment of the value added from analysis nudging. Nonetheless, that finding of superior accuracy does gives us confidence that the NCEP FNL product is a good basis for the generation of FDDA target fields, especially near the surface, and that nudging layers near the surface might lead to improved MPAS simulations if it can be done without disrupting the evolution of the simulated planetary boundary layer (PBL). This sort of disruption is why we have avoided nudging in the PBL for air-quality modeling purposes. We see this issue as a prime candidate for future work, but outside the scope of this effort.

Regarding sub-comment (a), we assume this suggests a comparison of MPAS-A with WRF. The physics parameterizations we use in WRF to support CMAQ modeling were designed specifically for air-quality assessment at a finer spatial scale than we used here. For this work, we were constrained to the physics options already available in MPAS-A. As a separate effort, we have begun work to add those special physics options to MPAS-A and will be applying them at 12-km horizontal resolution for comparison to WRF.

Regarding sub-comment (b), we have no intentions to develop a product in competition with CMAQ. Limited-area models will continue to have appropriate applications in both meteorological and air-quality simulations.

Comment: I am struck by the gross similarity between FDDA target and MPAS with FDDA in Figures 2-8. It would be good to see plots where the differences between FDDA target and MPAS with FDDA were presented.

Response: We added difference plots to Figs. 2-5 (now Figs. 2-3 and 6-7) and updated the discussion to describe the results. Adding difference plots to the vertical cross section and wind vector figures has proven to be difficult. We hope the additions we made will be sufficient. If not, additional time will be required for software modifications and new graphics preparation.

Comment: Can you give some theoretical reasoning for choosing the time-scale forcing to be around 55 minutes – it seems to be a bit on the strong side. The weaker forcing for q is around 9 hours. Maybe this exists in the WRF version.

Response: The time scale we adopted in MPAS is the same as the default value in WRF. The theoretical reasoning comes from Stauffer and Seaman (1990) where they equate that nudging time scale to that of meteorological phenomena at the meso-$\alpha$ spatial scale. This is now explained in the revised manuscript at the end of section 2.3. Our use of a longer time scale for moisture nudging in MPAS comes mainly from our experience with WRF where stronger nudging created too much cloudiness, too little downward shortwave radiation and degraded evaluation statistics at the surface. As shown in this work, weaker moisture nudging produces slightly better model accuracy.

Comment: Pg 3 Can the paper be a bit more specific on the details of how the NCEP FNL data is created?

Response: The exact process used by NCEP is continually being modified as their base modeling platform evolves and as new observational data become available (e.g.,

GOES-Next). The cited web reference for the NCEP FNL product provides a "Documentation" tab where various aspects of the NCEP FNL product are discussed. The availability of this information is indicated in the revised manuscript.

Comment: Throughout the paper the resolution of the NCEP FNL data is described in terms of 1-degree resolution which is around 100km. Given that MPAS resolution is described in terms of km can you mention the FNL data resolution in km to avoid the readers needing to do the conversion themselves.

Response: 1 degree of longitude is variable in physical length depending on latitude. Thus, it is not possible to describe the 1 x 1-degree resolution in terms of specific physical length. But the revised manuscript now mentions that the spatial resolution of the NCEP FNL product approximates that of the coarse portion of the MPAS mesh used in this study.

Comment: What is the timestep of the MPAS model?

Response: We used a timestep of 150 seconds as indicated in Table 1.

Comment: Can you use Figure throughout and not have a mixture of Figure, Fig. and (Fig. 9) on pages 6-10?

Response: With one exception, the original manuscript used the standard convention of "Figure" when starting a sentence and "Fig." otherwise. That exception on page 7, line 5 has been corrected.

Comment: Pg 9 line 11 RMSA -> RMSE

Response: This correction has been made.

Comment: Pg 9 line 30 remove dot: For January 2013

Response: That period was intended to be a comma. This has been corrected.

Comment: Pg 26 and Pg 27 Figures 7 and 8 need to include hyphen between 500 and

300 hPa

Response: These corrections have been made.

Comment: The plots in Figure 16 do not print out well. Can the units be included in the graph?

Response: Figure 16 shows mass values over time scaled to their starting value. Thus, no units are appropriate for these graphs. These graphs could be separated into individual figures for "total moist air mass", "total water vapor mass" and "total dry air mass", thus making the graphs larger and easier to see in print. This seems to be a print-versus-web formatting issue and we will defer to the editor as to whether Figure 16 needs to be expanded into three separate figures.

---

## Author Comment (AC2) · 6 Apr 2018

Comment: 1) As the focus of the manuscript is nudging as a DA method, the authors should extend the introduction section with the focus being more on nudging and less on air quality.

Response: We have extended the introduction to better describe the utility of nudging versus other data assimilation methods for the diagnostic purposes of this work. We are not really advancing the science of nudging but only extending its use to the MPAS

modeling platform. Therefore, we did not believe a full review of the development of nudging techniques was appropriate or necessary. The discussion regarding air quality modeling and the problems associated with the use of differing models and grid nesting for global-scale assessment is intended to explain the motivation for our use of MPAS. As can be seen in the first comment from Reviewer #1, there is some concern in the air-quality modeling community that our work is intended to create a replacement for CMAQ, which it is not.

Comment: 2) The implementation of the nudging algorithm and the setup of the experiment is well documented in the manuscript. However, there are some issues with the experiment setup which in my opinion are important to discuss: a) The observations assimilated into the model state are synthetic observations generated from NCEP analysis fields. As the implementation does work as expected, the nudged simulations closely resemble the input data. However, one major advantage of the nudging algorithm is to allow for a free simulation [of] the atmospheric processes corresponding to the model physics while being (temporally) constraint to the observations (or the target field in this case). With respect to the experiment design, the authors should add more discussion (and probably analysis) on this aspect. The experiment without PBL assimilation seems to be a natural starting point for this.

Response: We interpret this comment to say we should try to reduce or eliminate nudging between the times when target data are available to investigate the model's ability to provide a free simulation of atmospheric processes when guidance is lacking. The halting or gradual ramping down of analysis nudging has been applied elsewhere for the purposes of initializing prognostic simulations. However, this work has a necessary focus on diagnostic simulation. If we have misinterpreted this comment, we welcome clarification from the reviewer.

Comment: b) The setup of the experiment is based on the truth being represented by the target field. The synthetic observations are drawn from this field in a homogeneous and dense fashion. With this setup, it is not difficult to achieve an accordance of the

model with the target and the result will not strongly depend on the nudging coefficient(s). However, in reality, observations of the truth are heterogeneous, sparse and rare. Therefore, finding a reasonable and balanced setup which produces sound estimates of the atmospheric states is much harder especially with respect to the temporal availability of observations. Do the authors intend to do such experiments/simulations in the future? The authors should also provide results of sensitivity experiments with respect to the nudging coefficient.

Response: We agree that with analysis nudging it is easy to achieve an accordance of the model with the target fields by simply increasing the nudging coefficient. We also understand that nudging based on observational data (a.k.a., "obs nudging") is more difficult due to the heterogeneous and sparse nature of the observations, but that it also can provide additional accuracy with respect to the true state of the atmosphere. We intend to develop a method for "obs nudging" in MPAS-A, but that is beyond the scope of this initial effort.

Regarding sensitivity experiments with respect to the nudging coefficient, we had already conducted those using one-tenth, one-fifth and one-half nudging strength for all nudged variables. However, including those results here would result in an exceedingly long paper. We hope to submit a follow-on paper showing those sensitivity results in detail once this paper has demonstrated a proper facility for analysis nudging in MPAS-A.

Comment: 3) The authors should provide plots containing information on the analysis increments from the nudging with respect to its spatial and temporal variability.

Response: We have developed a new plot showing the temporal variation of simulated and target Ïť values and the nudging term for Ïť at layer 28 positioned over Research Triangle Park, NC (our laboratory location) during January 2013. That plot, which is Fig. 4 in the revised manuscript, shows a rather strong perturbation of Ïť near 0000 UTC on 18 January 2013. We also developed spatial plots, one focused on North
America to show detail, and a second showing the entire global domain for layer 28 at 0000 UTC on 18 January 2013. These spatial plots have been added as Fig. 5 of the revised manuscript. A short discussion of the results shown in Figs. 4 and 5 is included in the main text.

Comment: Page 2 Line 17: Please provide a reference for the Voronoi mesh.

Response: We have added a reference to Du, Q., V. Faber and M. Gunzburger (1999) Centroidal Voronoi Tessellations: Applications and Algorithms. SIAM Review Vol. 41, No. 4, pp.637-676.

Comment: Page 2 Line 23: Is the mesh really unstructured?

Response: No, on second thought it is not, and we thank the reviewer for pointing out the error in our original terminology. We now describe the mesh accurately as a centroidal Voronoi tessellation.

Comment: Page 6 Line 4: "show" instead of "shows"

Response: This correction has been made.

Comment: Page 7 Line 3: I am not able to comprehend what the authors want to say with this sentence.

Response: The sentence contained a typographical error where the word "if" should have been "of". We trust this correction will allow a proper comprehension.

Comment: Page 8 Line 29: It would rather say "are larger" instead of "are much larger"

Response: This correction has been made.

Comment: Page 9 Line 11: "RMSE" instead of "RMSA"

Response: This correction has been made.

Comment: Page 10 Line 2: "than" instead of "that"

[Figure]

Response: This correction has been made.

Comment: Discussion on Figures 10 to 13: The shape of the bottom and top diagram differ mostly by amplitude. Does this behavior arise from the fact that a majority of observations is coming from the CONUS region? Please elaborate on this.

Response: Yes, the CONUS region does have a high density of observations compared to the rest of the globe. This spatial concentration of observations probably does explain the temporal correlation of the bottom and top diagrams. However, finer model resolution over the CONUS and regional variation in the quality of observations complicate the matter. We were not able to draw firm conclusions as to why the line deflections are so similar for temperature and humidity (previously Figs. 10-13, now Figs. 12-15), but not so much for wind speed (previously Figs. 14-15, now Figs. 16-17). But we agree that this behavior needs to be mentioned and have included some discussion on the matter at the top on page 7 in the revised manuscript.
* * *

---

## Referee Report (RR1)

The manuscript "Adding Four Dimensional Data Assimilation by Analysis Nudging to the Model for Prediction Across Scales - Atmosphere (Version 4.0)" provides a description of the implementation of a nudging scheme into the MPAS model and provides results from an experiment testing this scheme by assimilating synthetic observations from a target field into the MPAS model state.

The manuscript is well-written and provides a detailed description of the implementations, experiments and results. Two of my three major issues as well as all of my minor comments with the first version of the manuscript have been addressed while the other major issue has been discussed to my satisfaction in the response by the authors. I find the manuscript to be much improved and I have no further issues regarding its publication. I therefore recommend "accept as is" for the manuscript.

---

## Author Response (AR2)

**Authors' response to reviewer and editor comments:**

Referee #1 recommended only that the resolution of figure 18 be improved before publication.

Referee #2 judged the previous manuscript to be "acceptable as is".

The first referee's recommendation is repeated in the editor's report and our responses below address that report. Individual editor comments are in bold text with associated responses immediately following in regular text.
* * *
**Please could you improve the resolution of figure 18 as indicated by referee #1.**

Response: Figure 18 has been divided into figures 18, 19 and 20 to provide improved resolution on each figure. The main text has also been modified to address the new numbering of figures.
* * *
**Please also consider the following additional comments:**

**- There are a number of web references in the main text (page 3 line 28, page 4 lines 12, 14, and 16, page 5 line 5, page 6 line 6). Is it possible to replace any of these with more permanent citations (e.g. to relevant Zenodo links if relevant), or with otherwise more complete citations?**

Response: The web reference at page 3 line 28 (and page 12 line 1) does have a DOI and that has been used in place of the previous reference. The other web references do not have associated Zenodo links or DOIs, but SourceForge (page 4 line 12), UCAR (page 4 line 14, page 5 line 5) and GitHub (Page 4 line 16) maintain these as permanent data repositories. The web reference at page 6 line 6 is the main portal to the MADIS, a meteorological observational database and data delivery system hosted by the U.S. Department of Commerce, NOAA. Hopefully that will also remain permanently, as we could find no citable publication to describe it.
* * *
**- Similarly can the source file referred to on page 10 line 15 be linked to the relevant model source code repository?**

Response: We assume the reference here is to page 10 line 10. The source file for the time integration module is included in the Zenodo code repository established for this project. This is indicated in the revised manuscript.
* * *
**- I find appendix B to be quite difficult to understand. Is it possible to add a bit more context to make the "Description" more accessible (e.g. what are the specific values indicating here)? Does the sentence starting "The basis for these changes ..." in appendix A also apply here?**

Response: The Description section of Appendix B is a direct quote from the MPAS GitHub web site where the primary MPAS developer at NCAR describes the nature of the problem. The sentence starting "The basis for these changes ..." in appendix A does also apply to the corrections described in appendix B. We added this sentence to appendix B and removed text from the Action section that was thus made redundant. We also edited the Action section (now titled Corrective Action) to improve comprehension.
* * *
**- Figures 2, 3, 6, "Difference" in the lower panels is a bit ambiguous here (although identified in the captions).**

Response: We tried using the label "FDDA Target - With FDDA", but it was too long. We hope the more detailed captions will suffice.
* * *
**- Figures 12-17, the blue lines are not identified in the captions.**

Response: The blue lines are identified in the revised captions.
* * *
In addition to the changes described above, we also made the following minor changes:

- We added a period after the "1" in the section 1 title.

- We updated the availability date for the MPAS users' guide on page 5 line 4.

- We replaced "FORTRAN" with "Fortran" in appendix A and appendix B to be consistent with the main text and modern convention.

- We corrected a missing opening parenthesis in the legend for the bottom graph in Figure 17.

[revised manuscript text omitted]

---

## Author Response (AR3)

**Authors' response to topical editor comment:**

**Please indicate more clearly, with a full reference, that the "description" in Appendix B is a direct quote from the GitHub link provided on the opening line of Appendix B.**

Response: Appendix B has been rewritten to more clearly indicate that the "description" is a direct quote. That quote is now uniquely formatted in italics and a full reference is made to it. Hopefully, the new entry in the list of references is in the proper format. We were not sure how to refer to a GitHub commit. The subsequent description of the corrective actions taken was also rewritten to make it more distinct from the quoted information.

[revised manuscript text omitted]